# Stepwise taming of triplet excitons via multiple confinements in intrinsic polymers for long-lived room-temperature phosphorescence

Liang Gao [1,3], Jiayue Huang[1,3], Lunjun Qu [1,3], Xiaohong Chen[1], Ying Zhu[1], Chen Li[1], Quanchi Tian[1], Yanli Zhao [2] ✉ & Chaolong Yang [1] ✉

Polymeric materials exhibiting room temperature phosphorescence (RTP) show a promising application potential. However, the conventional ways of preparing such materials are mainly focused on doping, which may suffer from phase separation, poor compatibility, and lack of effective methods to promote intersystem crossing and suppress the nonradiative deactivation rates. Herein, we present an intrinsically polymeric RTP system producing long-lived phosphorescence, high quantum yields and multiple colors by stepwise structural confinement to tame triplet excitons. In this strategy, the performance of the materials is improved in two aspects simultaneously: the phosphorescence lifetime of one polymer (9VA-B) increased more than 4 orders of magnitude, and the maximum phosphorescence quantum yield reached 16.04% in halogen-free polymers. Moreover, crack detection is realized by penetrating steam through the materials exposed to humid surroundings as a special quenching effect, and the information storage is carried out by employing the Morse code and the variations in lifetimes. This study provides a different strategy for constructing intrinsically polymeric RTP materials toward targeted applications.

Long-lived organic room-temperature phosphorescence (RTP) materials have been applied in illumination display[1], bioimaging[2], data encryption[3], and information anticounterfeiting[4] owing to their excellent biocompatibility, large Stokes shifts, and economical efficiency[5–7]. However, the intrinsic spin-forbidden transition from singlet to triplet excitons and the fast nonradiative decay of triplet excitons significantly decrease their luminous performance[8]. Various attempts have been made to prepare RTP systems based on two conditions: one is maximizing the intersystem crossing (ISC) from singlet to triplet states by introducing heteroatoms or heavy atoms to provide efficient spin-orbit coupling (SOC)[9,10]; the other is

restricting nonradiative transitions (collision, vibration, or rotation) in a rigid environment through crystal engineering, aggregation, or host-guest doping[10–12]. Organic afterglow luminous materials originating from polymer systems have received considerable attention because of their low cost, good biocompatibility, and simplicity of modification[13,14]. Doping phosphors with a polymer matrix is a classical approach for preparing polymeric RTP[12–16]. In particular, polyvinyl alcohol (PVA) matrices containing numerous dopants, including polymer@PVA[17], small molecule@PVA[12,18–22], and carbon dot@PVA[23], have been used to establish hydrogen-bonding networks (Fig. 1a).

[1]School of Materials Science and Engineering, Chongqing University of Technology, Chongqing 400054, China. [2]School of Chemistry, Chemical Engineering and Biotechnology, Nanyang Technological University, 21 Nanyang Link, Singapore 637371, Singapore. [3]These authors contributed equally: Liang Gao, Jiayue Huang, Lunjun Qu. ✉e-mail: zhaoyanli@ntu.edu.sg; yclly2013@cqut.edu.cn

**Fig. 1 | Doping and intrinsic polymerization systems based on polyvinyl alcohol matrix. a** Structures employed to prepare room-temperature phosphorescence by doping into polyvinyl alcohol matrix. **b** Strategy of constructing polymer-based room-temperature phosphorescence utilizing multiple confinements.

Furthermore, phosphor groups copolymerized with matrix monomers such as acrylic acid (AA)[24], acrylamide (AM)[25], styrene sulfonic acid/sodium[26,27], and vinyl pyridine (VPy)[28] have been developed to produce a range of ambient RTP copolymers (Supplementary Figs. 1–3) with superior features (Supplementary Table 1). In these systems, covalent bonds with high bonding energies are introduced to enhance the stability and weatherability of the materials when serving as bonding units of the molecules[29–31]. Compared to the case in hydrogen-bonded materials, the problem of phase separation in doped materials is significantly decreased[32]. Nevertheless, RTP has been achieved using a single- or one-way promotion of the phosphorescence process. These samples employ fewer covalent bonds, especially covalently crosslinked networks, resulting in faster nonradiative deactivation and poor anti-interference ability of the system. Several studies have demonstrated the utilization of quick-click reactions between boric acid and PVA to manufacture borate ester RTP under ambient conditions[33–36]. The aryl boronic acid employed in these systems is required to improve the processability and toughness of the frame, but this trait is detrimental when used to confine the regional movement of the cells. The high mobility necessarily increases the motion within the structure, thereby potentially causing a dramatic release of undesired nonradiative transition processes when added to RTP polymers[37]. Therefore, through structural confinement with

covalent crosslinking and hydrogen bonding, reasonable taming or protection of the radiation process of triplet excitons is promising for the development of RTP copolymers[11,33].

Herein, we present a stepwise methodology of three-level confinement: copolymerization of phosphors (chromophores) into poly(vinyl acetate) (PVAc) molecular chains to construct the primary confinement copolymer, and hydrogen bonding networks from PVA by alcoholysis of PVAc as the secondary confinement. The rapid reaction between PVA and boric acid produces an intrinsically crosslinked RTP copolymer and consequently achieves tertiary confinement (Fig. 1b). With an increase in stepwise structural confinement, the lifetime enhances from 14.3 μs to 256.5 ms, with the nonradiative rate constant of RTP ($k_{nr}^{P}$) restricted to 3.9 s$^{-1}$ from polymerization (primary confinement) to crosslinking (tertiary confinement). The phosphorescence quantum yield reaches 16.04% in halogen-free polymers at room temperature as the structure is progressively confined. The experimental data demonstrate that the ISC of the system is gradually enhanced, and the theoretical calculation results indicate that the vertical excitation energy of the triplet state is enriched after crosslinking. The luminous performance of the materials is enhanced in two ways simultaneously. Finally, we develop a strategy for information encryption and establish a method for detecting microcracks (<2 mm) under humid conditions by utilizing the water-sensitive features of PVA.

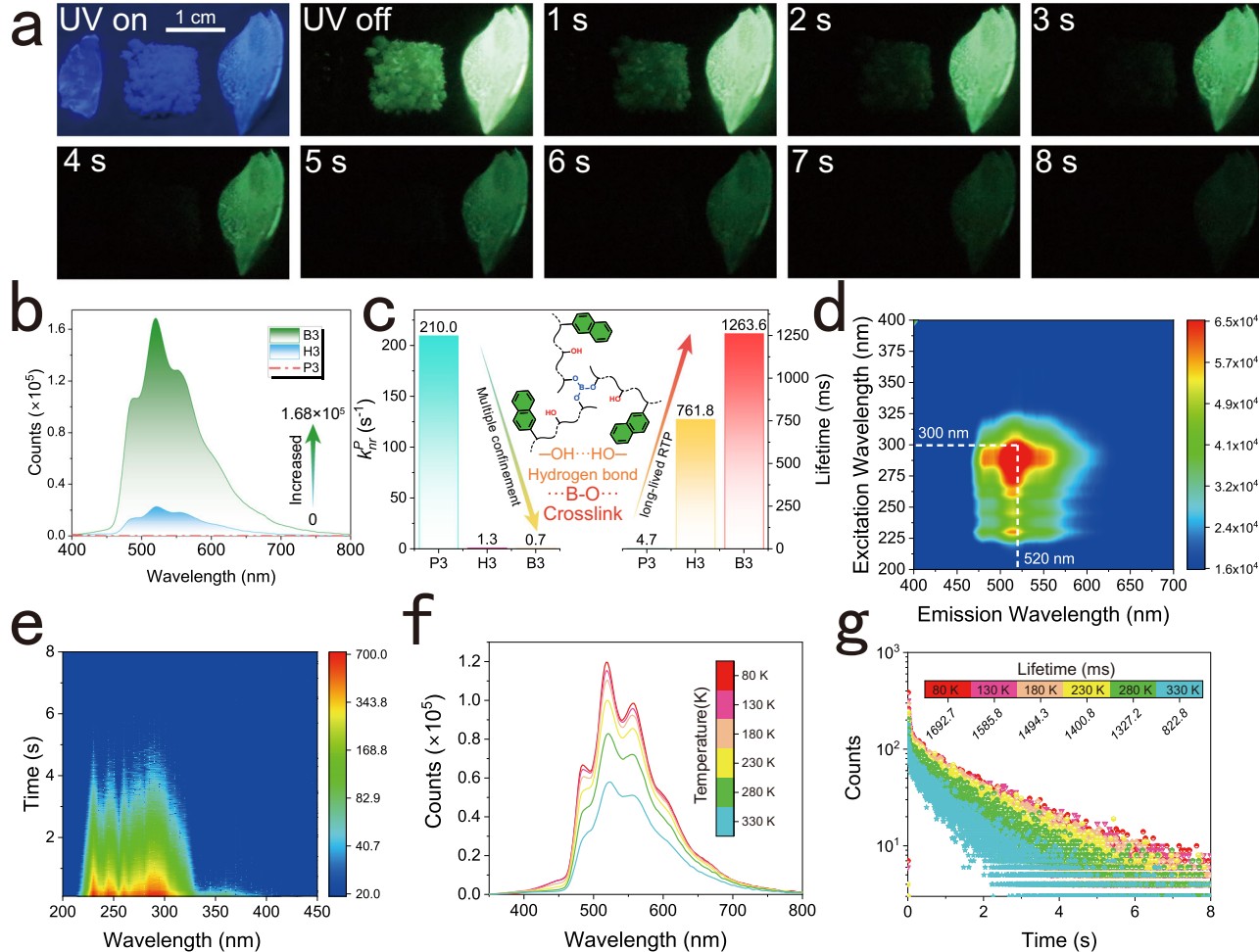

**Fig. 2 | Stepwise structural confinement from P3 to H3 and to B3.**
**a** Phosphorescent decay behavior under 254 nm at ambient conditions (from left to right: P3, H3, and B3). **b** RTP emission spectra under the same condition with $\lambda_{ex} = 300$ nm and delay = 5 ms. **c** Trend of $k_{nr}^{P}$ and fitted lifetime from the decay spectra under room temperature, where the illustration is a sketch of the crosslinked structure of B3. **d** Two-dimension excitation-dependent emission spectra of B3. **e** Two-dimension time-resolution excitation spectra of B3 ($\lambda_{em} = 520$ nm). **f, g** Temperature-dependent phosphorescence (**f**) and decay (**g**) spectra of B3 at 520 nm excited by 300 nm μF lamp.

## Results

The stepwise molecular confinement strategy was demonstrated by comparing the characteristics of the following materials: vinyl acetate (VAc) was copolymerized with the phosphor segment 2-vinyl naphthalene (2VN) to construct the primary confinement copolymer (P), and the PVA copolymer (H) was obtained by alcoholysis for secondary confinement. Tertiary confinement of the intrinsic copolymer (B) was realized by further crosslinking with boric acid. Theoretical calculations suggest that covalent crosslinking within the polymer, rather than weak interactions, is responsible for the RTP emission. Four different feeding ratios of 2VN were explored and the enhanced luminescence was observed with increasing system rigidity (Supplementary Table 2).

Particularly in the third group (Fig. 2a), the afterglow was poorly apparent at room temperature in the primary confinement copolymer (P3), and bright RTP emission was observed during alcoholysis (H3), which was further amplified by crosslinking as a tertiary confinement (B3). The afterglow of B3 lasted for more than 10 s after excitation with a 254 nm UV lamp (Supplementary Movie 1). The photophysical characteristics of P3, H3, and B3 were determined under identical conditions (Fig. 2b). Among the three, the increase in RTP emission expectancy is the most obvious, with the intensity changing from virtually invisible (P3) to $1.68 \times 10^5$ (B3) and exhibiting a yellow-green afterglow. Their lifetime increases from 4.70 ms to 1263.60 ms (Fig. 2c;

Supplementary Figs. 4, 5), and the phosphorescence quantum yield increases from 0.06% (P3) to 10.17% (B3) (Supplementary Table 3), which is a leap from primary to tertiary confinement. These results show that stepwise confinement is an effective method for improving RTP performance.

Next, the excitation-dependent emission spectra and time-resolution excitation spectra (TRES) of B3 indicate that the RTP emission is not excitation-dependent and is mainly centered at 520 nm (Fig. 2d, e). The Commission Internationale de l´Eclairage (CIE) coordinates of B3 and H3 are (0.33,0.53) and (0.33,0.52) respectively (Supplementary Fig. 4). The fluorescence emission wavelength is listed in Supplementary Table 3, which is located in the blue region of corresponding CIE coordinates. These changes illustrate the stability of the photoluminescence spectra. The fluorescence lifetime ($\tau_F$) is 50.8 ns in P3 and decreases to 38.8 ns after alcoholysis (H3) with the increase in ISC rate constant ($k_{isc}$) from $1.2 \times 10^4 \, s^{-1}$ to $6.4 \times 10^5 \, s^{-1}$. Then, $\tau_F$ and $k_{isc}$ reached 42.0 ns and $2.4 \times 10^6 \, s^{-1}$ in B3, respectively. The change in fluorescence quantum yields ($\Phi_F$) is consistent with $\tau_F$ (Supplementary Table 3), meaning that a higher triplet state (T) is produced through ISC and the non-radiative path of T is restricted as the rigidity of the polymeric structure increases. Similar enhancement effects and constant RTP emission were observed in the other three groups (Supplementary Figs. 5–17; Supplementary Movies 2–4). The decrease in intensity

(Fig. 2f) and lifetime (Fig. 2g) indicate a standard RTP during the heating operation, which continued from 80 K. With the absence of hydrogen bonds and covalent crosslinked bonds in P3, the lifetime of P3 sharply decreases to 9.8 ms at 230 K. This transformation of the triplet exciton is consistent with nonradiative decay, which gradually occupies a dominant position in the relaxation process (Supplementary Fig. 18).

To further investigate the role of the stepwise structural confinement strategy on the phosphorescence behavior, we estimated the rate constants associated with the excitons during decay, where $k_{isc}$ and the nonradiative decay rate constant ($k_{nr}^P$) exhibited opposing tendencies (Supplementary Table 3). For instance, in P3, H3, and B3, $k_{nr}^P$ decreases by three orders of magnitude from $2.1 \times 10^2 \, s^{-1}$ to $7.1 \times 10^{-1} \, s^{-1}$, accompanied by a steady rise in $k_{isc}$. The nonradiative deactivation channel of the triplet exciton was continuously suppressed, and phosphorescence increased gradually, as indicated by a progressive reduction in $k_{nr}^P$ during the change from P to H to B[33]. The above results demonstrate that the stepwise confinement strategy promotes ISC, while compressing the nonradiative process of triplet excitons and finally activating RTP.

The reasons for these events were discussed through structural evidence. Following alcoholysis of the four groups based on 2VN, the number of average molecular weight were listed in Supplementary Table 4. The four groups of B1–B4 after crosslinking within powder X-ray diffraction (XRD) exhibit a typical amorphous structure, and the decomposition behavior of P3 is consistent with that of PVAc.

In hydroxyl-containing samples, there is a step in the cross-linked sample around 400 °C. Combined with the infrared characteristics of 300-400 °C in thermogravimetric-infrared (TG-IR) spectra, this step behavior is caused by the collapse of the cross-linked structure (Supplementary Fig. 19). Considering the poor solubility after boric acid crosslinking, the changes in P3 with primary confinement and H3 with secondary confinement in [1]H NMR were studied (Supplementary Fig. 20). The chemical shift of methyl hydrogen at the ester group is 4.78 ppm, and that of the vinyl group at 5.35−6.01 ppm in 2VN becomes nearly undetectable in H3 accompanied by the evident hydroxyl shift at 4.22−4.67 ppm. Meanwhile, H3 additionally gathers aromatic ring H vibrations at approximately 6.5−8.0 ppm (inset image of Supplementary Fig. 20b)[38]. Certified polymeric systems contain both phosphorus and hydroxyl groups.

Corresponding changes were also found in the spectra obtained by Fourier-transform infrared (FTIR) and X-ray photoelectron spectroscopy (XPS). The stretching vibration of C = O at 1727 cm$^{-1}$ almost disappeared in H3 and B3 compared to that in P3. The hydrogen bonding vibration at 3276 cm$^{-1}$ was sharper in H3 than that in B3 and was undetectable in P3 (Fig. 3a). In addition, the stretching vibration belonging to the B-O bond at 1287 cm$^{-1}$ was also detected[39]. These findings indicate that P3 did not provide hydrogen bonds or covalent crosslinking as strong as those originating from H3 and B3, based on the almost complete alcoholysis process. The redshift of the hydrogen bonds also led to the formation of a new hydrogen-bond network after crosslinking[12]. The other three groups underwent similar

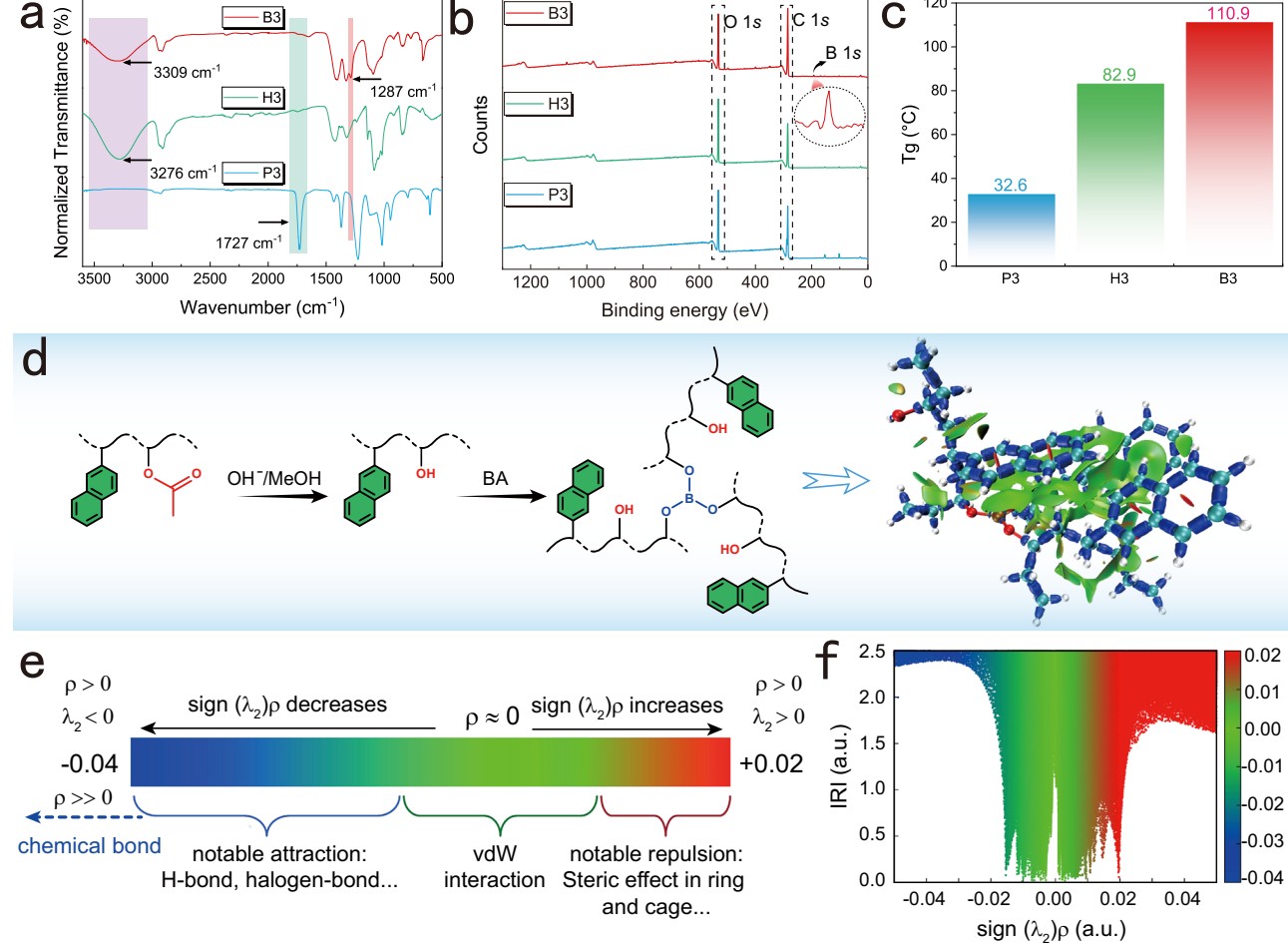

**Fig. 3 | Structural characterization and theoretical calculations.**
a–c Comparison of structural properties of FTIR (a), XPS (b), and $T_g$ (c) before (P3) and after (H3) alcoholysis and after crosslinking (B3). d Synthetic routes and visualization of interactions in crosslinked structures. e Color projections of sign ($\lambda_2$)$\rho$ on the interaction region indicator (IRI) isosurfaces and chemical significance. f Distribution of different interactions in B3 via IRI isosurfaces. a.u. Atomic units.

transformations (Supplementary Fig. 21). For XPS, the energies of C = O and C-O in O 1 $s$ of P3 are 531.9 and 533.1 eV, respectively. In comparison, the three types were fitted to 284.8 (C-C), 286.4 (C-O), and 288.9 eV (C = O) from the C 1 $s$ (Supplementary Fig. 22). Contrarily, the C 1 $s$ of the alcoholysis and crosslinked polymers were observed for C-O of 286.1 and 286.3 eV (for H3 and B3, respectively) and for C-C of 284.6 and 284.8 eV (for H3 and B3) respectively, equating to 532.3 eV in both O 1 $s$ of C-O (Supplementary Fig. 22) and the B-O from B3 fitted at 192.2 eV (Fig. 3b; Supplementary Fig. 23)[40,41]. Furthermore, the glass transition temperatures ($T_g$) of the polymers progressively increased. No hydrogen bonding and no crosslinking network were observed in P3, where the $T_g$ is 32.6 °C. In H3, hydrogen bonding is present, and the $T_g$ is 82.9 °C, whereas that of B3 reaches 110.9 °C via a covalent crosslink network (Fig. 3c). The enhanced effect in the other three sets of experiments showed a resembled learning (Supplementary Fig. 24). This transformation supports the impact of stepwise confinement of the triplet exciton to achieve its taming of the triplet exciton.

In addition to cross-linked covalent bonding, noncovalent interactions occur between structures or molecular atoms, such as hydrogen bonds, dipole–dipole interactions, spatial repulsion, and dispersion across a broad range of bonding energies[42]. These interactions are buried in the bonding network and are challenging to visualize using conventional methods[42,43]. Therefore, an interaction region indicator (IRI) was introduced to indicate weak

interactions. To theoretically calculate the structure according to the synthesis method, a simplified model was used because of the complex structure of the polymeric system (Fig. 3d). The spatial coordinates are listed in Supplementary Data. The structure was imported into Multiwfn for IRI analysis with the optimization results, and the graph was plotted to obtain a projection of the IRI on the interaction mapping of sign($\lambda_2$)$\rho$, where $\rho$ represents the electron density and $\lambda_2$ denotes the second largest eigenvalue of the Hessian matrix of the electron density[44]. Bonded and non-bonded interactions are distinguished by the sign of $\lambda_2$. The bonding area (blue), van der Waals (vdW) forces (green), and spatial effects (red) in the ring structure are all displayed in sign($\lambda_2$)$\rho$ in that order[42] (Fig. 3e). Weak interactions are located in the blue and green intervals (−0.04 to 0.02).

In the 2VN system, weak interactions in this region from primary (P) and secondary (H) confinement were not evident (Supplementary Fig. 25) despite having a large potential difference and area in the electrostatic potential (ESP) of the primary confinement system[45] (Supplementary Fig. 26). These ESPs are focused on the O atom of the ester group and dispersed between the hydrogen atoms of the carbon chain and the aromatic ring (Supplementary Fig. 27). The positive potential (red) centered on the hydrogen of the hydroxyl group and the negative potential (blue) is concentrated in the O atom of the hydroxyl group in the presence of hydroxyl groups

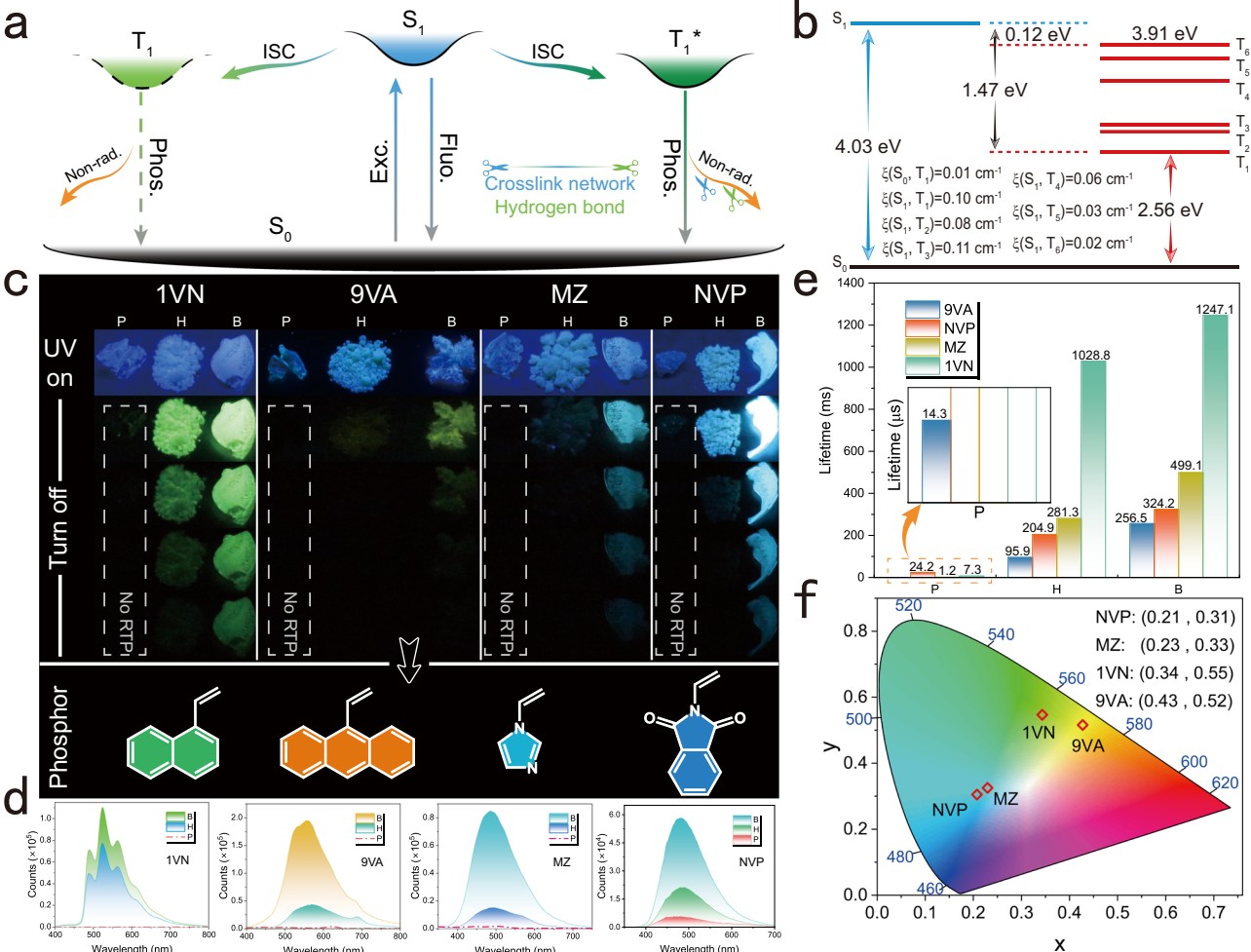

**Fig. 4 | Verification of the suitability of the strategy for different structures. a** Mechanism of stepwise confinement in a polymeric compound. **b** Vertical excitation energy of the optimized and simplified model B3. **c** Phosphorescence decay behavior under 254 nm at ambient conditions in different systems. **d, e** Phosphorescence spectra (**d**) and lifetime (**e**) excited by 300 nm (enlargement unit is μs). **f** CIE coordinates of each example after crosslinking. Fluo. fluorescence, Phos. phosphorescence, Exc. excitation, ISC intersystem crossing, Non-rad. nonradiative decay.

after alcoholysis (Supplementary Fig. 27). The possibility of generating hydrogen bonds is substantially higher than that for the hydrogen-free ester group, and the $T_g$ is in perfect agreement (Fig. 3c). Consequently, the emergence of hydrogen bonding during alcoholysis confines the previous flexible system and compresses the nonradiative pathways. Next, the IRI and ESP of B (crosslinked) were much higher than those of the two preceding systems (H and P) (Fig. 3f; Supplementary Figs. 25, 27)[46,47]. Overall, the weak interaction buried between molecules achieves the primary confinement of excitons, hydrogen bonds realize secondary confinement based on the previous interaction, and covalent bonds are introduced to establish tertiary confinement. Such interactions significantly compress the polymeric nonradiative pathways (Fig. 4a) and enrich the vertical excitation energy of triplet excitons to boost the RTP (Fig. 4b; Supplementary Fig. 25).

The suitability of this strategy was extended to four systems through subsequent applications (Fig. 4c): 1-vinyl naphthalene (1VN, Supplementary Movie 5), 9-vinyl anthracene (9 VA, Supplementary Movie 6), vinyl imidazole (MZ, Supplementary Movie 7), and *N*-vinyl phthalimide NVP (Supplementary Movie 8). First, these four systems show similar enhancement trends to 2VN (Supplementary Figs. 28–53): phosphorescence before alcoholysis is extremely limited and is almost invisible to the naked eye. The phosphorescence behavior gradually increased as the emission wavelengths became yellow-green (1VN-B:520 nm), yellowish (9VA-B:558 nm), and blue (NVP-B:482 nm; MZ-B:486 nm) (Fig. 4d), and the phosphorescence lifetimes increased 1 or 4 orders of magnitude from P to B (Fig. 4e; Supplementary Figs. 28–31), respectively. This resulted in multicolor coverage of blue, green, and yellow in the CIE coordinates (Fig. 4f). The maximum phosphorescence quantum yield was 16.04% for this strategy in NVP with explicit crosslinking (Supplementary Table 5).

As shown in Supplementary Fig. 32, –BO bonds were detectable at approximately 1284–1288 cm⁻¹, and the fitted B-O binding energy is 192.3 eV (NVP, MZ and 9 VA), and 193.4 eV (1VN) respectively, while hydrogen bonds were located at approximately 3325–3328 cm⁻¹. After alcoholysis, the C = O carbonyl group around 1729–1730 cm⁻¹ disappeared, indicating the completion of the alcoholysis process. Sharper peak-shaped hydrogen bonding occurred[39] in the range of 3281–3285 cm⁻¹. As $k_{isc}$ and $T_g$ increased and $k_{nr}^P$ gradually decreased, the structure was progressively confined between them (Supplementary Figs. 33, 34; Supplementary Table 5). Powder XRD exhibited typical amorphous features after crosslinking (Supplementary Fig. 35), as well as RTP behavior in the variable temperature spectra (Supplementary Figs. 36–39). Theoretical analysis of these four systems using a simplified model showed that the IRI, ESP, and vertical excitation energy changes after crosslinking exhibited the same characteristics as those of 2VN (Supplementary Figs. 40–47). Notably, in MZ and NVP, the existence of N and O heteroatoms in the phosphor caused the negative potential of the system to expand in the structure of the phosphorescent unit (Supplementary Figs. 44–47), thereby enriching the weak interactions within the system.

Owing to its high strength, light weight, as well as superior thermal and electrical insulation abilities, polymer materials are widely employed in plastic packaging, coatings, textiles, biomedicine, and other industrial constructions[48]. From manufacturing to usage, these materials may be affected by mechanical stress and environmental conditions. Polymer chains can be damaged by internal stress and external influences (such as corrosive chemicals, heat, ultraviolet rays, and mechanical shocks)[48]. Microcracks begin developing continuously with the growth of the materials as caused by environmental factors including temperature, humidity, chemicals, and radiation[48,49].

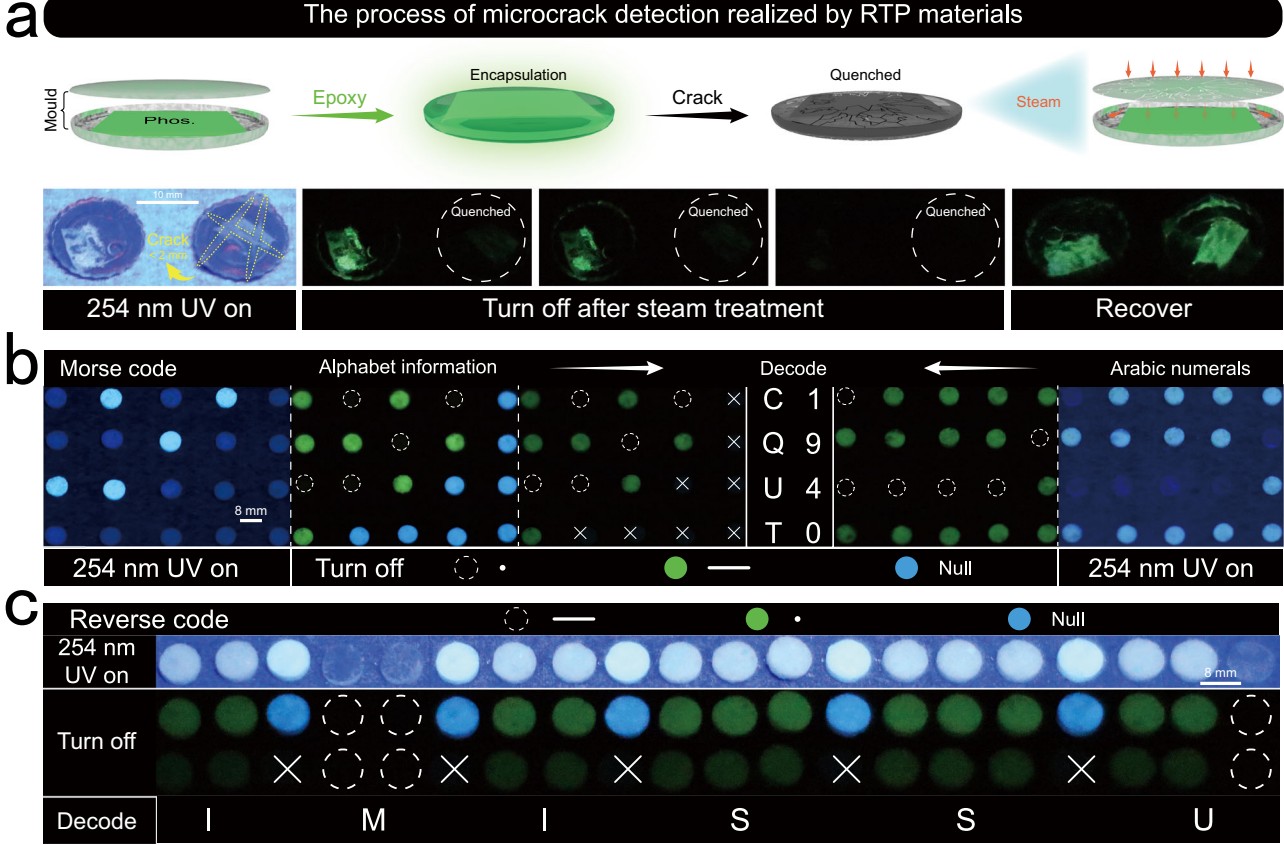

**Fig. 5 | Applications of intrinsic polymeric RTP materials. a** Microcrack detection in humid environments. **b**, **c** Analysis of the encryption and decryption process of the Morse code (**b**) and flipping the compilation process (**c**).

It creates a passageway for the entry of corrosive liquids like water, oxygen, and other substances, which makes thermo-mechanical properties of the materials deteriorated. Currently, the main techniques used to detect microcracks in materials are non-destructive testing (e.g., visual testing and computed tomography scanning)[50]. These methods are often restricted by the discontinuity of the cracks, the depth of the defects, the shape of the materials itself, and technical barriers to the use of professional equipment[48,51]. Therefore, it is crucial to develop straightforward and practical testing procedures. Herein, we designed a method for detecting cracks in humid environments using RTP polymers based on the water-sensitive features of PVA. This was achieved by encapsulating the crosslinked RTP materials in epoxy resin and creating artificial cracks in the epoxy resin to simulate cracks (<2 mm) generated by external factors, such as stress concentration, in a real application scenario (Fig. 5a; Supplementary Fig. 54). The fumigation process was carried out for 14 h to ensure that water vapor penetrated the materials through the surface cracks, adsorbed by the polymers, and then allowed to cool to room temperature. The control samples were subjected to atmospheric conditions throughout the experiment. After UV turn-off, the phosphorescence of the cracked samples was significantly quenched, and that of the control group lasted several times. As seen in Supplementary Fig. 55a, the phosphorescence of the experimental group (Microcrack) was nearly wiped out in comparison to the control group. The phosphorescence lifetime of the experimental group was insufficient for fitting, while the control group possessed a lifetime of approximately 690 ms (Supplementary Fig. 55b, c). Finally, the quenched RTP reappeared after activation in an oven at 65 °C, suggesting that the quenching effect of water vapor is effective (Supplementary Movie 9). We experimented again when both sets of materials were dried (Supplementary Fig. 55d–f), and the phosphorescence lifetime was improved by a certain amount. Especially, the lifetime of cracks was recovered to 682.5 ms.

Additionally, a method for information encryption with the Morse code was developed using the differences in lifetimes between samples. First, the horizontal line —— in the Morse code was represented by the long-lived H3, the dot • was defined by 9VA-H with a short lifetime, and the in-between NVP-H represented the interference information symbol ×. A clear difference was observed between the CQUT, where the alphabet was applied, and numbers 1940, which used the Arabic numeric table and had no interference information. When the excitation source was turned on, the horizontal line and interference information were hidden. When the excitation source was switched off, 9VA-H disappeared. Compared to the numeric table, the alphabet still contained interference information at this time, indicating a higher encryption level (Supplementary Movie 10). With the disappearance of the interference information, the Morse code table decryption information yielded CQUT and 1940 (Fig. 5b; Supplementary Fig. 56). The flexibility of the encryption method also facilitates the encryption of flipped information. For example, the long lifetime was used to represent the point information, the horizontal line information was represented by the short lifetime, and the original interference information was used to describe the interruption. In total, the information was used to yield the information encryption of the lengthy statement I MISS YOU (Fig. 5c).

## Discussion

In summary, the PVAc copolymer (P) used in this strategy exhibits weak interactions comparable to those of PVA. Its insufficient hydrogen-bonding ability and low $T_g$ prevent the inhibition of the nonradiative deactivation pathway of the triplet exciton at room temperature, thus limiting the release of RTP. However, the PVA copolymer (H) created by alcoholysis depends on the hydrogen bonding between the chain segments to achieve secondary confinement of the phosphor, and the prepared polymers can release

RTP after alcoholysis with desirable luminescence properties. Multicolor RTP emissions with blue, green, and yellow were achieved by adjusting the amount of copolymerized phosphor. Benefitting from the quick reaction between the boronic acid and alcohol hydroxyl groups, the B-O-C covalent bonding crosslinked network and the unreacted hydrogen bonds in the structure further compress the excitonic $k_{nr}^{P}$, resulting in about 4 orders of magnitude decrease (the lifetimes increased from 14.3 µs to 256.5 ms) and achieving a phosphorescent lifetime of 1.26 s with the enhancement of $k_{isc}$ at room temperature. With the intervention of heteroatoms in the crosslinked NVP-B, a phosphorescent quantum yield of up to 16.04% was achieved. The suitability of this strategy was demonstrated in terms of ESP and IRI through theoretical analyses and experiments. Finally, the strategy was applied to the detection of microcracks in humid environments that encapsulate the crosslinked RTP material in epoxy resin, as well as to information storage with the Morse code using the differences in lifetime between the samples. This strategy expands the application area of purely intrinsic and polymeric RTP materials, providing a design strategy for the development of advanced RTP materials.

## Methods

### Synthesis of P1 (mol. VAc:2VN = 300:1)

VAc (0.22 mol, 18.48 g), methanol (20 mL), 2VN (0.72 mmol, 0.11 g), and 2,2'-azobisisobutyronitrile (AIBN, 1 wt. %, relative to total weight of VAc and 2VN) were added to a Schlenk tube for vacuum degassing and argon (Ar) circulation for 5 times. The mixture was then heated to 65 °C for 48 h and poured into deionized water (600 mL) to obtain the white precipitate. The solid was dried at 65 °C under vacuum to obtain the polymerization product (before alcoholysis), which was denoted by P.

### Synthesis of H1

P1 (8 g) was dissolved in methanol (90 mL) at 45 °C with mechanical stirring. Stirring was continued for 2 h after the addition of NaOH methanol solution (2 mL, 5%), followed by adding NaOH methanol solution (1.5 mL, 5%) for 1 h. Finally, the temperature was raised to 65 °C for 2 h to obtain a white turbid solution. The solution pH was adjusted to ≤ 7 using dilute hydrochloric acid. The samples were filtered and washed with methanol for 5 times. The alcoholysis product H was obtained by vacuum drying at 65 °C.

### Synthesis of B1

H1 (0.5 g) was dissolved in deionized water (10 mL) at 100 °C on a magnetic heating agitator. Then, the prepared boric acid solution (50 mL, 0.5 mol/L) was added under stirring while the solution was hot. The solution changed from turbid to transparent. The viscous solids were collected and washed thrice with deionized water. After vacuum drying at 65 °C, the crosslinked sample B was obtained.

The polymerization products P2, P3, and P4 (before alcoholysis) were obtained under the same conditions as P1 except for the feed proportion. H2, H3, and H4 were prepared by alcoholysis of the related polymerization product. The crosslinked products B2, B3, and B4 were prepared using the homologous alcoholysis product H under the same conditions as B1.

### P2 (mol. VAc:2VN = 500:1)

VAc (0.22 mol, 18.48 g), 2VN (0.43 mmol, 0.07 g).

### P3 (mol. VAc:2VN = 700:1)

VAc (0.22 mol, 18.48 g), 2VN (0.31 mmol, 0.05 g).

### P4 (mol. VAc:2VN = 1000:1)

VAc (0.22 mol, 18.48 g), 2VN (0.22 mmol, 0.03 g).

The polymerization products P (mol. VAc:Phos = 700:1) with different phosphors (Phos) were prepared by copolymerization of 1VN (0.31 mmol, 0.05 g), 9 VA (0.31 mmol, 0.06 g), NVP (0.31 mmol, 0.05 g), or MZ (0.31 mmol, 0.03 g) with VAc (18.48 g, 0.22 mol). The process for preparing the alcoholysis (H) and crosslinking (B) polymers was the same as those of H1 and B1, respectively.

## Microcracks

Microcracks were achieved by encapsulating the crosslinked RTP material B3 in the epoxy resin in the mold and creating artificial cracks in the epoxy resin to simulate the cracks generated by external factors and to simulate the stress concentration in a real application scenario. Samples with cracks were fumigated in a steam environment. After heating and softening, the crack width can be appropriately increased (≤2 mm) and infiltration treatment can be done, and the total fumigation process can be maintained for 14 h. The solution was then cooled to room temperature. The samples in the control group were placed in an atmospheric environment with epoxy resin and then compared with the RTP to observe the degree of quenching of the samples.

## Information encryption

The sample was molded at room temperature in a circular mold with an inner diameter of 8 mm. The alphabet: horizontal line —— was represented by the long-lived H3, the dot • was represented by 9VA-H with a short lifetime, and the in-between NVP-H represented the interference information symbol ×. The Arabic numeric: horizontal line —— was represented by the long-lived H3, and the dot • was represented by P3 with a short lifetime. The long statement: horizontal line —— was represented by the short-lived 1VN-P, the dot • was represented by 1VN-H with a long lifetime, and the in-between NVP-H represented the interference information symbol ×.

## Data availability

All the data supporting the findings of this study are available within the article and its supplementary information files and from the corresponding author upon request.

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

## Acknowledgements

This work was financially supported by the National Natural Science Foundation of China (22275025 and 21875025, C.L.Y.), the Innovation Research Group at Institutions of Higher Education in Chongqing (CXQT19027, C.L.Y.), the Chongqing Talent Program, the Science and Technology Project of Banan District, the Innovation Support Plan for the Returned Overseas of Chongqing (cx2020052, C.L.Y.), the Open Fund of Guangdong Provincial Key Laboratory of Luminescence from Molecular Aggregates (2021-kllma-03, C.L.Y.), and the Agency for Science, Technology and Research (A*STAR) Singapore through Its Manufacturing, Trade and Connectivity (MTC) Individual Research Grant (M22K2c0077, Y.L.Z.).

## Author contributions

L.G., J.Y.H., and C.L.Y. conceived and were responsible for the experiments. L.G., J.Y.H., X.H.C., Y.Z., and Q.C.T. synthesized the copolymers and performed the photoluminescence measurements. J.Y.H. and C.L. completed the preparation of the application. L.J.Q. performed theoretical calculations. L.G., Y.L.Z., and C.L.Y. performed the data analysis and wrote the manuscript. All authors contributed to the final version of the manuscript.

## Competing interests

The authors declare no competing interests.
