## [Peer Review File · Nature Communications]

Stepwise taming of triplet excitons via multiple confinements in intrinsic polymers for long-lived room-temperature phosphorescenceREVIEWER COMMENTS

Reviewer #1 (Remarks to the Author):

This work demonstrated a polymeric RTP system through the stepwise methodology of three-level confinement. Authors have shown that the covalently cross-linked network formed due to the B-O bond formation increases the rigidity of the system along with hydrogen bonding between hydroxyl. Theoretical analyses and experiments are the better evidence to prove the feasibility of the strategy. Surprisingly, authors have applied it to the detection of microcracks. In addition, the authors have carefully summarized the development and main problems in the field of polymeric RTP research. The work contributes significantly to the area and is of sufficient novelty to warrant publication in Nature Communications. It will be of interest to a wide readership. However, I have a few minor comments that the authors may wish to consider.

- 1) In the manuscript, the alcoholysis polymer "H" released RTP, the authors should provide additional experimental data in the text to avoid the influence of the pure PVA matrix.
- 2) Why does the low electrostatic potential (Supplementary Figure 24) in the first three materials appear in the "P" system, while the latter two materials appear in the "B" system?
- 3) In the manuscript, On page 7. "The four groups of B1–B4 after crosslinking within X-ray diffraction (XRD) exhibit a typical amorphous structure". Does the author need to make some necessary analysis and explanation of TG and TG-IR (Supplementary Figure 17)?
- 4) In the manuscript, the XPS analysis of 1VN, MZ, 9VA and NVP should be provided to support the existence of B-O bond.
- 5) Why did the authors choose the ratio of 700:1 of NVP, 9VA, MZ and 1VN? Some explanation should be given in revised manuscript
- 6) In Supplementary Figure 18 and Supplementary Figure 51, the corresponding ¹³C NMR spectra should be provided to further confirm the structures of desired polymers.
- 7) Since polymeric room-temperature phosphorescence (RTP) materials reported in this manuscript, I suggest the authors to cite some additional related references. Such as Adv. Sci. 2022, 9, 2103402; Chem. Sci., 2023,14, 5177-5181, Chemical Engineering Journal 2022, 433, 134307. et al.

Reviewer #2 (Remarks to the Author):

In this work, Zhao et al reported a feasible and facile strategy to achieve long-lived polymeric room-temperature phosphorescence. Through structural confinement with covalent crosslinking and hydrogen bonding, the RTP lifetime and quantum yield of the material were improved simultaneously in halogen-free polymers. The results are meaningful for gaining ultralong lifetime RTP materials. Overall, this work is intriguing and this manuscript is worth of publication in Nature Communications after some minor revisions.

1. In Figure 2f, the phosphorescent lifetime of B3 at 330 K is 822.8 ms. How will the phosphorescence property change if the temperature further increase?

2. In Figure 5a, additional spectra change of quenched and control experiments needs to be provided, and the microcracks displays were too vague to identify, a scale needs to be added.
3. The legend of supplementary figure 26a needs to be revised.
4. There are some mistakes in the caption of Figure 4e "enlargement unit is ms".
5. How does the authors confirm that the afterglow of NVP, 9VA, MZ and 1VN comes from T-state luminescence? The authors need to provide some necessary explanations in revised manuscript.

Reviewer #3 (Remarks to the Author):

In this manuscript NCOMMS-23-22830-T, the authors describe a material design for polymeric systems to show very high room-temperature phosphorescence (RTP). While the manuscript contains interesting results, I do not think that is of that high importance to justify publication in a very high impact journal like Nature Communications. The overall presentation also needs major improvements. Below I summarize the main points that led me to my conclusion. Overall I do not recommend publication in Nature Communications.

1. I disagree with the first sentence of the abstract. I do not think that one can say that polymeric RTP systems are of central importance in organic optoelectronics.
2. The abstract does not contain any hint at the second demonstration in form of the encryption scheme, as presented in Figure 5b,c and according text elements.
3. I find the presentation of increasement factors (e.g. like 170,000-fold) a way to oversell scientific results. Here, the starting point is simply showing virtually no RTP, so clearly any reasonable RTP emission will be attributed with such a massive increase. I would suggest to use better ways to report the changes.
4. Figure 1 has two problems: in a) I am not sure what this material vs. timeline should demonstrate. This is not a review article, where this might be interesting to summarize. in b), I am not sure why the authors use leaves and flowers to indicate their chemical structure. It almost seems like an attempt to 'green wash' the approach. This is not needed. A more objective presentation would be favored.
5. Page 5, central section: It states that the fluoescence is in a range of 336 -338 nm, which would span only 2 nm. Please correct.
6. I am very confused about the materials and their naming. In the main manuscript, the units with number 3 (P3, H3, B3) are used. What are series 1 and 2? Later, further systems are named. Very confusing.
7. Overall, the discussion of the results is very superficial. A lot of numbers and ratios are discussed, but the scientific depths is missing. Try to focus on smaller material sets and discuss properly.

8. Figure 2e should contain the fluorescence band, but the x-axis features span from 225 to 325 nm. This cannot be the fluorescence, can it? There seems to be something wrong.

9. Page 7, first sentence 2nd paragraph: 'The mechanism of this study was ...' I cannot extract the meaning of this sentence.

10. On comment on the encryption scheme, presented in Figure 5b,c. These presentations are very often shown in various types of papers before and they always seem to be made more and more detailed and complex, why there is little new content.

11. The crack detection application is interesting, but discussed way to little. This would deserve a separate publication.

RESPONSE TO REVIEWERS' COMMENTS

Reviewer #1 (Remarks to the Author):

This work demonstrated a polymeric RTP system through the stepwise methodology of three-level confinement. Authors have shown that the covalently cross-linked network formed due to the B-O bond formation increases the rigidity of the system along with hydrogen bonding between hydroxyl. Theoretical analyses and experiments are the better evidence to prove the feasibility of the strategy. Surprisingly, authors have applied it to the detection of microcracks. In addition, the authors have carefully summarized the development and main problems in the field of polymeric RTP research. The work contributes significantly to the area and is of sufficient novelty to warrant publication in Nature Communications. It will be of interest to a wide readership. However, I have a few minor comments that the authors may wish to consider.

Response: We thank the precious time of the reviewer devoted to the reviewing process. We appreciate the reviewer for the evaluation of our manuscript. We have done our best to improve the manuscript according to the reviewer's suggestions.

1) In the manuscript, the alcoholysis polymer "H" released RTP, the authors should provide additional experimental data in the text to avoid the influence of the pure PVA matrix.

Response: Thanks for the good comment. At first, we characterized the PVA prepared in the laboratory (Lab) by infrared and gel permeation chromatography to confirm the structure information (Figure R1a). The strong IR vibration of hydroxyl groups can be seen at $2980\text{ cm}^{-1} \sim 3697\text{ cm}^{-1}$. The delayed spectra of Lab and commercial PVA were tested, and the very weak emission of them are located at 468 nm (Lab) and 472 nm (Commercial) respectively. Combined with the alcoholysis PVA copolymer (H) prepared after copolymerization with the phosphorescent units in the manuscript, the RTP emission wavelength is $\lambda_{em} \geq 490\text{ nm}$. Therefore, the phosphorescent unit is responsible for the strong RTP emission and PVA chains play an important role in the construction of the rigid system. The related data have been added in the manuscript.

Figure R1. (a) Infrared spectrum of PVA synthesized in laboratory (Lab) with the number average molecular weight $M_n = 11472$ Dalton; Delayed emission spectra with $\lambda_{ex} = 300\text{ nm}$ of synthesized in laboratory (b) and commercialized product purchased from Macklin (c).

2) Why does the low electrostatic potential (Supplementary Figure 24) in the first three materials appear in the “P” system, while the latter two materials appear in the “B” system?

Response: Thanks for your good question. First of all, it can be seen from their electrostatic potential (ESP) distribution that in the first three systems (2VN, 1VN and 9VA), the negative/low electrostatic potential (blue area) mainly comes from the contribution of the carbonyl group (Figure R2a-c) in polyvinyl acetate (PVAc), while in the latter two systems (NVP, MZ), the negative electrostatic potential comes from carbonyl group and heteroatoms such as N, O in phosphorescence unit (Figure R2d-e). After alcoholysis, the contribution of the carbonyl group is replaced by the hydroxyl group, part of the hydroxyl groups would be consumed after further crosslinking, and more phosphorescence units would be introduced (Figure R2f-g). Thus, the system only depending on hydroxyl group to provide negative electrostatic potential energy is obviously weaker than the system containing heteroatom and hydroxyl group. The related data have been added in the manuscript.

Figure R2. Distribution of ESP in polymerization of 2VN (a), 1VN (b), 9VA (c), MZ (d), NVP (e) and after cross-linked of MZ (f) and NVP (g). Negative potential (blue), positive potential (red), and neutral potential (white). The balls are the extreme point of regional potential (blue: minimum value; yellow: maximum value).

3) In the manuscript, On page 7. “The four groups of B1–B4 after crosslinking within X-ray diffraction (XRD) exhibit a typical amorphous structure”. Does the author need to make some necessary analysis and explanation of TG and TG-IR (Supplementary Figure 17)?

Response: Thanks so much for your good suggestion. The explanation for this part is as follows: The polymerized sample (before alcoholysis) P3 containing the carbonyl group matches the decomposition characteristic of PVAc. In hydroxyl-containing samples, there is a step in the cross-linked sample around 400 °C. Combined with the infrared characteristics of 300-400 °C in TG-IR. This step behavior should be caused by the collapse of the cross-linked structure. We have added the explanation to the revised manuscript on page 7.

4) In the manuscript, the XPS analysis of 1VN, MZ, 9VA and NVP should be provided to support the existence of B-O bond.

Response: Thanks for your good suggestion. According to your suggestion, we have carried out XPS analysis of the crosslinked polymers from 1VN, 9VA, MZ and NVP respectively. In the B 1s spectra, the fitted B-O binding energy is 192.3 eV (NVP, MZ and 9VA) and 193.4 eV (1VN). Combined with their IR spectra (Supplementary Fig. 30), B-O bonds were detectable at the range of 1284–1288 cm^{-1} and the glass transition temperatures (T_g) of the polymers progressively increased (Supplementary Fig. 32). These results provide sufficient evidence for the existence of the B-O bond. The related content was added to the revised manuscript and Supplementary Figure 30.

Figure R3. (a) X-ray photoelectron spectroscopy of crosslinked polymer prepared by copolymerization of different phosphorescence units. XPS high-resolution B1s spectra of NVP (b), 9VA (c), 1VN (d) and MZ (e).

5) Why did the authors choose the ratio of 700:1 of NVP, 9VA, MZ and 1VN? Some explanation should be given in revised manuscript.

Response: Many thanks for your good question. In the manuscript, a series of enhancement effects with 2VN as a phosphorescence unit after polymerization (P), alcoholysis (H) and cross-linking (B) were discussed at first. Among them, the phosphorescence properties under different ratios (300:1, 500:1, 700:1 and 1000:1) based on 2VN were characterized. In these four groups of different proportions, the lifetime and intensity of phosphorescence are gradually enhanced with the increase of the rigidity of the system. It demonstrates the flexibility of the strategy to certain circumstances. Therefore, this method has been utilized for several systems to broaden their possibilities of applications. Referring to the long-lived phosphorescence for the case of 700:1 in 2VN as the view of cost-saving, the same proportion was incorporated in NVP, 9VA, MZ and 1VN systems with an expectation that they would have produced similar findings.

6) In Supplementary Figure 18 and Supplementary Figure 51, the corresponding ^{13}C NMR spectra should be provided to further confirm the structures of desired polymers.

Response: Thanks for your good suggestion. According to your suggestion, additional ^{13}C NMR spectra were obtained from $\text{DMSO}-d_6$ and added to Figure R4 and Figure R5, respectively. The information in the corresponding Supplementary Figure 18 and Supplementary Figure 51 has also been modified. For more details, please refer to the revised Supplementary Information.

Figure R4 (Supplementary Figure 18). ^1H NMR spectra of P3 (a), H3 (b), and 2VN (c) obtained from $\text{DMSO}-d_6$. ^{13}C NMR spectra of P3 (d), H3 (e), and 2VN (f) obtained from $\text{DMSO}-d_6$.

Figure R5 (Supplementary Figure 51). ^1H NMR spectra of phosphors (a) and the copolymer after alcoholysis (b) obtained from $\text{DMSO-}d_6$. ^{13}C NMR spectra of different phosphors (c-f) and their copolymer after alcoholysis (g-j) obtained from $\text{DMSO-}d_6$.

7) Since polymeric room-temperature phosphorescence (RTP) materials reported in this manuscript, I suggest the authors to cite some additional related references. Such as Adv. Sci. 2022, 9, 2103402; Chem. Sci., 2023,14, 5177-5181, Chemical Engineering Journal 2022, 433, 134307. et al.

Response: Thanks for your good suggestion. Polymeric room-temperature phosphorescence (RTP) materials have thermoplasticity, flexibility and processability. A series of polymeric RTP was prepared using such thermomechanically plasticized, bright and long-lived RTP from these references. We have added these references to the revised manuscript, please see Ref. 15-16 and 36.

Reviewer #2 (Remarks to the Author):

In this work, Zhao et al reported a feasible and facile strategy to achieve long-lived polymeric room-temperature phosphorescence. Through structural confinement with covalent crosslinking and hydrogen bonding, the RTP lifetime and quantum yield of the material were improved simultaneously in halogen-free polymers. The results are meaningful for gaining ultralong lifetime RTP materials. Overall, this work is intriguing and this manuscript is worth of publication in Nature Communications after some minor revisions.

Response: We thank the precious time of the reviewer devoted to the reviewing process. We appreciate the reviewer for the positive evaluation of our manuscript. We have done our best to improve the manuscript according to the reviewer's suggestions.

1. In Figure 2f, the phosphorescent lifetime of B3 at 330 K is 822.8 ms. How will the phosphorescence property change if the temperature further increase?

Response: Many thanks for your good question. Starting at 300 K, the decay behavior of phosphorescence lifetime and intensity of B3 was explored. The spectra show that the phosphorescence intensity drops are nearly undetectable (Figure R6a) and the lifetime decreases from 844.2 ms to 22 ms at 450 K (Figure R6b). Combining these facts with the glass transition temperature ($T_g=383.9$ K) of the material, it can be concluded that the ability of movement from the chain segment continues to increase as the temperature rises, and the subsequent non-radiative process takes over the deactivation, which gradually suppresses the radiation pathway and leads to the decline of phosphorescence performance. Thus, it can be noticed that the inhibition power of cross-linking network between the glass transition temperature and room temperature may also allow materials to keep their specific phosphorescence characteristics.

Figure R6. Temperature-dependent phosphorescence (a) and decay (b) spectra of B3 at 520 nm excited by 300 nm μ F lamp.

2. In Figure 5a, additional spectra change of quenched and control experiments needs to be provided, and the microcracks displays were too vague to identify, a scale needs to be added.

Response: Thank you very much for your good question. The cracked sample has been pretreated by fumigation, followed by cooling down to room temperature. The control parts were exposed to ambient conditions during the experiment for this period. These two groups of phosphorescence were compared under the same conditions. As seen in Figure R7a, the experimental group's phosphorescence (Microcrack) was nearly wiped out compared to the control. The phosphorescence lifetime of the experimental group has been insufficient for fitting, while the control group possesses a lifetime of approximately 690 ms (Figure R7b,c). We experimented again shortly when both sets of materials had dried (Figure R7d–f), and the phosphorescence lifetime had improved by a certain amount, especially the lifetime of cracks, which had been recovered to 682.5 ms. This part was added to the revised manuscript and Supplementary Figure 53.

As concerns to the microcrack displays, we have redrawn the callouts around the cracks in Figure R8 and added a scale bar to the picture (bottom right). Figure 5 had been modified in the revised manuscript.

Figure R7 (Supplementary Figure 53). Phosphorescence spectra of control and microcrack groups after fumigation (a) and the recovery (d). Decay spectra and their lifetime of control and microcrack groups after fumigation (b, c) and the recovery (e, f)

Figure R8. Photos of control and microcrack groups under 254 nm UV lamp.

3. The legend of supplementary figure 26a needs to be revised.

Response: Thanks for your good comment. The modified figure is shown in Figure R9. The information in the corresponding Supplementary Figure 26a has also been modified. For more details, please refer to the revised Supplementary Information.

Figure R9 (Supplementary Figure 26a). Decay spectra and fitted lifetime of crosslinked at room temperature from 1VN.

4. There are some mistakes in the caption of Figure 4e “enlargement unit is ms”.

Response: Thanks for your question. The correct unit should be microseconds (μs), and the manuscript has been modified in the legend of Figure 4e. For more details, please refer to the revised manuscript.

5. How does the authors confirm that the afterglow of NVP, 9VA, MZ and 1VN comes from T-state luminescence? The authors need to provide some necessary explanations in revised manuscript.

Response: Thanks so much for your question. There is a specific redshift phenomenon and the emission peaks in their delayed spectra at room temperature differ from those in photoluminescence spectra. According to the formula of $E=1240/\lambda$, it is clear that the energy level of the delayed emission peak is lower compared to the energy level of the photoluminescence peak. In a series of variable temperature spectra that we measured in Figure R10, while comparing to the temperature-dependent spectra in Figure 2f-g, Supplementary Figure 16 and Supplementary Figure 35-37, we noticed that the delay spectra and lifetime reduced with a rise of temperature. The afterglow is generated from T-state luminescence, which is a typical feature of phosphorescence at room temperature.

Figure R10. Temperature-dependent phosphorescence of 1VN-H (a), 9VA-H (b), MZ-H (c) and NVP-H (d) excited by 300 nm μ F lamp. Decay spectra and fitted lifetime from 1VN-H (e), 9VA-H (f), MZ-H (g) and NVP-H (h) at different temperatures.

Reviewer #3 (Remarks to the Author):

In this manuscript NCOMMS-23-22830-T, the authors describe a material design for polymeric systems to show very high room-temperature phosphorescence (RTP). While the manuscript contains interesting results, I do not think that is of that high importance to justify publication in a very high impact journal like Nature Communications. The overall presentation also needs major improvements. Below I summarize the main points that led me to my conclusion. Overall I do not recommend publication in Nature Communications.

Response: We would like to thank the reviewer for reviewing our manuscript. We appreciate the review's acknowledgement that our manuscript contains some interesting results. The traditional ways of preparing long-lived polymeric RTP are focused on doping and employ fewer covalent bonds, especially covalently crosslinked networks, resulting in faster non-radiative deactivation and poor anti-interference ability of the system. In this work, we have demonstrated an intrinsically polymeric RTP system by stepwise structural confinement. The luminous performance of the materials is enhanced as the structure was progressively confined. This study outlines a straightforward, highly effective, and applicable strategy for the fabrication of long-lived polymeric RTP.

Compared with previous work, the interactions are buried in the bonding network, which are challenging to visualize using conventional methods. We have combined theoretical simulations to visualize the interaction where the weak interaction buried between molecules achieves the primary confinement of excitons, hydrogen bonds realize secondary confinement, and covalent bonds are introduced to establish tertiary confinement. Such interactions significantly compress the polymeric non-radiative pathways and enrich the vertical excitation energy of triplet excitons to boost the RTP. This breakthrough represents a major progress in this field and reveals the enhancement mechanism of pure long-lived polymeric RTP.

Finally, the strategy was applied to the detection of microcracks and information storage. This strategy expands the application area of pure intrinsic polymeric RTP materials and provides a design concept for the customizable platform, which opens up new possibilities for their applications in various optoelectronic fields.

We have done our best to improve the manuscript according to the reviewer's suggestions. We sincerely hope that the revised manuscript is suitable for publication.

1. I disagree with the first sentence of the abstract. I do not think that one can say that polymeric RTP systems are of central importance in organic optoelectronics.

Response: Thanks so much for your question. Emissive materials are the functional components of some technologies (Acc. Chem. Res. 2022, 55, 1573-1585), and the present organic optoelectronic materials are utilized extensively in organic light-emitting diodes (OLED), sensing, display, and other disciplines (Angew. Chem. Int. Ed. 2017, 56, 16207-16211). Nevertheless, small molecule organic optoelectronic materials have limitations, such as expense and demands for

doping when employed (Adv. Mater. 2020, 32, 2003911). Therefore, there are several potential applications for cheaper non-doping or solution-process polymeric materials in the development of flexible optoelectronic devices (Nat Mater. 2022, 21, 338-344). According to your good suggestion, to avoid any potential misunderstandings, we have revised the description of this statement to illustrate the flexibility of the polymer materials. The revised content is as follows: "Polymeric materials exhibiting room temperature phosphorescence (RTP) have a promising application potential in the fabrication of flexible optoelectronic devices".

2. The abstract does not contain any hint at the second demonstration in form of the encryption scheme, as presented in Figure 5b,c and according text elements.

Response: Many thanks for your good suggestion. We oversimplified the abstract in original manuscript. According to your suggestion, we have added corresponding contents in revised abstract: "Moreover, crack detection is realized by penetrating steam through the materials exposed to humid surroundings as a special quenching effect, and information storage is carried out by employing the Morse code and the variations in lifetimes."

3. I find the presentation of increasement factors (e.g. like 170,000-fold) a way to oversell scientific results. Here, the starting point is simply showing virtually no RTP, so clearly any reasonable RTP emission will be attributed with such a massive increase. I would suggest to use better ways to report the changes.

Response: Thanks so much for your suggestion. The data from 9VA, with a lifetime of 14.3 μ s before alcoholysis and 256.5 ms following cross-linking--a real >17,000-fold increase-- is the case that we presented in the manuscript and is truthful and valid. Because of the poor interactions in pre-alcoholysis materials (9VA-P), phosphorescence is virtually undetectable. With the enhancement of the confinement effect resulting from alcoholysis and continued cross-linking, the emission of RTP has been significantly improved, demonstrating the viability of the strategy. Although the phosphorescence performance of other groups is better compared to that of 9VA, the enhancement of "increment factors" in this case is most significant when compared to other groups. According to reviewer's remainder, we have revised this expression in the introduction and conclusion of the manuscript. Please see the revised manuscript for more details.

4. Figure 1 has two problems: in a) I am not sure what this material vs. timeline should demonstrate. This is not a review article, where this might be interesting to summarize. in b), I am not sure why the authors use leaves and flowers to indicate their chemical structure. It almost seems like an attempt to 'green wash' the approach. This is not needed. A more objective presentation would be favored.

Response: Thanks for your comments. The first question is as follows: Figure 1a focuses on an introduction of current research on PVA doping, providing the inspiration for the paper's main

idea. In these doped materials, small molecules, polymers, and carbon dots have been established. However, there are few reports on PVA materials that are intrinsic polymers. "Polyvinyl alcohol (PVA) matrices containing numerous dopants, including polymer@PVA, small molecule@PVA, and carbon dot@PVA, have been used to establish hydrogen-bonding networks (Fig. 1a)." The manuscript includes information of these associated contents on page 2 as well.

As for the second question, Figure 1b is to demonstrate the design idea of this manuscript. To avoid using straightforward and monotonous statements about the structures used in the article, as well as to make it simpler to understand the design concept, different colors of leaves were used to replace the polymer materials before (PVAc, blue) and after (PVA, green) alcoholysis, and flowers represent chromophore (phosphorescence units) on polymer segments. The corresponding chemical structures can be found in the synthetic route of Supplementary Information. This is not an attempt to 'green wash'. Finally, we have also made the necessary changes to Figure R11 (Fig. 1) to clarify the topic by following the reviewer's recommendations.

Figure R11 (Fig. 1). Doping and intrinsic polymerization system based on polyvinyl alcohol matrix. a) Structures were employed to prepare room-temperature phosphorescence by doping into polyvinyl alcohol matrix. b) Preparation and crosslinking structure of intrinsic room-temperature phosphorescence polymer.

5. Page 5, central section: It states that the fluorescence is in a range of 336 -338 nm, which would span only 2 nm. Please correct.

Response: Thank you very much for reminder. To avoid unnecessary misunderstand, we have modified this part as follows: “The fluorescence emission wavelength is listed in Supplementary Table 2”.

6. I am very confused about the materials and their naming. In the main manuscript, the units with number 3 (P3, H3, B3) are used. What are series 1 and 2? Later, further systems are named. Very confusing.

Response: Thanks for your question. In this manuscript, by employing 2VN as a phosphorescence unit, four series of enhancement phenomena during polymerization (P), alcoholysis (H), and cross-linking (B) were explored. Among them, series 3 (700:1) was studied in detail and the copolymer of series 1 (300:1), series 2 (500:1) and series 4 (1000:1) were presented in the Supplementary Information since their improving result and phosphorescence performance are similar to series 3. Series 1 and 2 mentioned by the reviewers in the comment are also introduced on pages 5 and 8 in the manuscript, and more information has been put into Supplementary Figures 4-15, 19 and Supplementary Movies 2-4. The feasibility of this strategy was then confirmed through the addition of various phosphorescence units. By comparisons between the horizontal and vertical axes, it is discovered that as the confinement effect grows, the performance increases. The following table marks the names of different samples and their corresponding shortened forms.

Table R1. Names of series 1 to 4 and different phosphorescence units.

VAc : Phos	2VN				1VN	9VA	MZ	NVP
	300:1	500:1	700:1	1000:1				
Polymerization (P)	P1	P2	P3	P4	1VN-P	9VA-P	MZ-P	NVP-P
Alcoholysis (H)	H1	H2	H3	H4	1VN-H	9VA-H	MZ-H	NVP-H
Cross-linking (B)	B1	B2	B3	B4	1VN-B	9VA-B	MZ-B	NVP-B

7. Overall, the discussion of the results is very superficial. A lot of numbers and ratios are discussed, but the scientific depths is missing. Try to focus on smaller material sets and discuss properly.

Response: Many thanks for your question. We apologize for bothering the reviewers when we utilized series 3 to highlight this improvement for better clarity. As noted in the above query, we went into a great detail about series 3. The performance of the phosphorescence improves with the enhancement of confinement, based on the photophysical data. We further explained that

the improvement in phosphorescence performance is caused by the co-promotion of K_{isc} and K_{nr} by harvesting a series of transition constants buried in the photophysical data. Next, from the perspective of structural facts, it continues to clarify the origin of these observations. Originally, NMR, XPS, and IR were employed to confirm the structures. Then, with an enhancement of the confined result, the glass transition temperature determined by DSC displayed an upward tendency. Therefore, the structural changes provided are the basis of this series of improvements. Only by analyzing these data in detail can we excavate the mechanism behind the experimental phenomenon.

Series 1, 2, and 4 subsequently demonstrated similar results to series 3. The trapping effect is similarly favorable in the copolymers developed by changing other phosphorescence units, demonstrating the flexibility of the strategy. Weak interaction force is a non-intuitive virtualization effect that is challenging to measure with accuracy through experimental methods. As a result, we conducted theoretical calculations to demonstrate the interactions between structures and chemical bonds by interaction region indicator (IRI) and verified the systems. The weak interaction is directly connected to the structure's potential distribution, and thus the electrostatic potential (ESP) distribution is created as well. It is also clarified that the change of structure leads to the weak interaction between the differences in the distribution of electrostatic potential. Additionally, the difference in apparent photophysical properties is induced by the covalent cross-linking network. We corresponded the physical phenomena to the structural changes and verified the enhancement effect of the structural changes through theoretical simulations. We discussed and analyzed the reasons step by step. Finally, we provided different strategies for constructing intrinsically polymeric RTP materials and expanding their applications.

Overall, while the methods employed may seem straightforward, they entailed substantial effort and extensive refinement to ensure optimal results. As suggested, we have carefully revised manuscript with deeper discussions based on the experimental data obtained.

8. Figure 2e should contain the fluorescence band, but the x-axis features span from 225 to 325 nm. This cannot be the fluorescence, can it? There seems to be something wrong.

Response: Thanks so much for your suggestion. Figure 2e is the two-dimension time-resolution excitation spectra (TRES) of phosphorescence ($\lambda_{em}=520$ nm), and the legend of the manuscript has been corrected. This figure consists of multiple phosphorescence decay spectra under different excitation wavelengths. Combined with Figure 2d, it shows that there is no excitation-dependence in this material. The microsecond lamp used in the test has a frequency of 0.125 Hz with a source trigger delay and a cycle of 8 s. The fluorescence lifetime is on a nanosecond scale from Supplementary Figure 13 and would not interfere with the test.

9. Page 7, first sentence 2nd paragraph: 'The mechanism of this study was ...' I cannot extract the meaning of this sentence.

Response: Thanks for your comments. We sincerely apologize for the inconvenience brought to the reviewer. The photophysical characteristics of the materials were discussed in the previous paragraphs. The structural information of materials is then described, and the relationship between this phenomenon and structure is investigated to further explain the mechanism of stepwise enhancement. This sentence was rewritten in the revised manuscript as follows: Next, the reasons for these events were discussed through structural evidence.

10. On comment on the encryption scheme, presented in Figure 5b,c. These presentations are very often shown in various types of papers before and they always seem to be made more and more detailed and complex, why there is little new content.

Response: Thank you for your suggestion. Phosphorescence materials are widely used in a variety of areas (Chem 2023, DOI: 10.1016/j.chempr.2023.05.023), including volatile organic compound (VOC) detection (J. Am. Chem. Soc. 2022, 144, 6107-6117), biological imaging (Nat. Commun. 2022, 13, 186), temperature sensors (Angew. Chem. Int. Ed. 2023, 62, e202302751), and information encryption (Nat. Commun. 2022, 13, 7423). Information encryption is acknowledged by numerous researchers as a traditional application method. When developing innovative application strategies, it should not only investigate whether the physical features match strategies, but also take the applicability of traditional means into account. In our manuscript, we are not confined to the application of data encryption, but we also explore innovative uses. In Figure 5a, the sensitivity of hydrogen bonds to water was employed to establish microcrack detection.

11. The crack detection application is interesting, but discussed way to little. This would deserve a separate publication.

Response: Thanks for your good suggestion. According to your suggestion, additional spectral changes of quenched and control experiments are provided in Figure R12. As seen in Figure R12a, the phosphorescence from the experimental group (Microcrack) was nearly wiped out in comparison to the control group. The phosphorescence lifetime of the experimental group has been insufficient for fitting, while the control group possesses a lifetime of approximately 690 ms (Figure R12b, c). We experimented again when both sets of materials had dried (Figure R12d-f), and the phosphorescence lifetime had improved by a certain amount, especially the lifetime of cracks, which had been recovered to 682.5 ms. This part was added to the revised manuscript and Supplementary Figure 53. In this work, we have explored the application potential of the developed phosphorescent systems in crack detection. Combining the application demonstration in the manuscript would give a better integrality of the work. Thanks for the understanding.

Figure R12 (Supplementary Figure 53). Phosphorescence spectra of control and microcrack groups after fumigation (a) and the recovery (d). Decay spectra and their lifetime of control and microcrack groups after fumigation (b, c) and the recovery (e, f).

REVIEWER COMMENTS

Reviewer #1 (Remarks to the Author):

The authors have made substantial revisions according to reviewers' comments and all concerns have been addressed. I recommend its publication in Nature Communication with its current version.

Reviewer #2 (Remarks to the Author):

The reviewers' concerns have been fully addressed by the authors. The revised manuscript can now be accepted for publication as is.

Reviewer #3 (Remarks to the Author):

This review is related to the revised manuscript of NCOMMS-23-22830A. In the following, I will relate only to my (Reviewers 3) comments of my original review and the related replies of the authors. Overall, the authors have submitted an elaborate rebuttal letter which is accompanied with a revised main text.

The authors have addressed all the comments in the original review with detailed replies, but many major items have not been resolved, rather the reply gives reason, why the original presentation is like it is. Some examples:

1. Re: Comment 1: I still must disagree. The presented materials (RTP materials) are not utilized in OLED technology for good. With connecting just generally to organic optoelectronics, I fear that the materials of this study (and similar materials) are suggested to become a central contribution of organic optoelectronics. I do not see that. RTP materials have much more a use as photonic subsystems - this may seem as a subtle difference, but it clearly should not be confused.

2. Re: Comment 3: I understand that the factor 17000 is real, but the wording make it suggest more than it is. I would prefer a increase/decrease by a certain factor (or by more than 4 orders of magnitude), or giving the respective values from and to which the systems changes. (I am not happy with X-fold, ...)

3. Re: Comment 4: While there are some subtle changes to the figure, I still do not like it to have this cartoon like impression, which misses the scientific depths one would expect here.

Many of the other comments are replied to in much detail, but the actual changes are comparable small with respect to the original manuscript. Therefore overall, I am assessing that this manuscript is not suited for a broad readership of Nature Communications, it still lacks clarity and is in my judgement not general enough.

Response to Reviewers' Comments

Reviewer #1 (Remarks to the Author):

The authors have made substantial revisions according to reviewers' comments and all concerns have been addressed. I recommend its publication in Nature Communication with its current version.

Response: We thank the precious time of the reviewer devoted to the reviewing process. Thanks very much for the recommendation of publication.

Reviewer #2 (Remarks to the Author):

The reviewers' concerns have been fully addressed by the authors. The revised manuscript can now be accepted for publication as is.

Response: We thank the precious time of the reviewer devoted to the reviewing process. Thanks very much for the recommendation of publication.

Reviewer #3 (Remarks to the Author):

This review is related to the revised manuscript of NCOMMS-23-22830A. In the following, I will relate only to my (Reviewers 3) comments of my original review and the related replies of the authors. Overall, the authors have submitted an elaborate rebuttal letter which is accompanied with a revised main text.

The authors have addressed all the comments in the original review with detailed replies, but many major items have not been resolved, rather the reply gives reason, why the original presentation is like it is. Some examples:

Response: We really appreciate the reviewer for the evaluation of our manuscript. We have done our best to improve the manuscript according to your good suggestions.

1. Re: Comment 1: I still must disagree. The presented materials (RTP materials) are not utilized in OLED technology for good. With connecting just generally to organic optoelectronics, I fear that the materials of this study (and similar materials) are suggested to become a central contribution of organic optoelectronics. I do not see that. RTP materials have much more a use as photonic subsystems - this may seem as a subtle difference, but it clearly should not be confused.

Response: Thanks for your useful comments. To avoid any misunderstanding, we have deleted the redundant statement of fabricated optoelectronic devices in Abstract section. The revised content is as follows: "Polymeric materials exhibiting room temperature phosphorescence (RTP) show a promising application potential."

2. Re: Comment 3: I understand that the factor 17000 is real, but the wording make it suggest more than it is. I would prefer a increase/decrease by a certain factor (or by more than 4 orders of magnitude), or giving the respective values from and to which the systems changes. (I am not happy with X-fold, ...)

Response: Thanks so much for your good suggestion. The factor of "X-fold" from this manuscript has been deleted and replaced with the "orders of magnitude " and their respective values. The revised content is as follows:

- 1). On page 1 of the abstract the ">17,000-fold", the "10-fold to 17,000-fold" from page 12, and "17,000-fold" located on page 17 are all replaced with orders of magnitude;
- 2). At the end of page 4 the "> 17,000-fold" as well as the "269-fold" and "170-fold" of page 6 were deleted and employed the actual numerical changes.
- 3). At the end of page 17 the "200-fold" was deleted directly.

3. Re: Comment 4: While there are some subtle changes to the figure, I still do not like it to have this cartoon like impression, which misses the scientific depths one would expect here.

Response: Thank you very much for kind reminder. According to your constructive comments,

the “leaves and flowers” in Figure 1a were deleted, the co-polymer materials of PVAc and PVA are shown in their corresponding chemical structures as presented below.

Figure 1. Doping and intrinsic polymerization system based on polyvinyl alcohol matrix. a) Structures were employed to prepare room-temperature phosphorescence by doping into polyvinyl alcohol matrix. b) Strategy of constructing intrinsic polymer room-temperature phosphorescence utilizing multiple confinements.

4. Many of the other comments are replied to in much detail, but the actual changes are comparable small with respect to the original manuscript. Therefore overall, I am assessing that this manuscript is not suited for a broad readership of Nature Communications, it still lacks clarity and is in my judgement not general enough.

Response: We would like to thank the precious time of the reviewer for reviewing the manuscript. Among other comments from the previous version, you put forward good suggestions on many details of the original manuscript. In previous comments, you suggested that the crack detection application is interesting, but the discussion is too little. Thus, we added a detailed description of crack detection application in the revised version. To further clarity the importance of the crack detection, in this revision, we rearranged the introduction of microcracks and rewrote this part in the manuscript on page 14. The detailed contents are as follows:

“Owing to its high strength, light weight, as well as superior thermal and electrical insulation abilities, polymer materials are widely employed in plastic packaging, coatings, textiles, biomedicine, and other industrial constructions.⁴⁸ From manufacturing to usage, these

materials may be affected by mechanical stress and environmental conditions. Polymer chains can be damaged by internal stress and external influences (such as corrosive chemicals, heat, ultraviolet rays, and mechanical shocks).⁴⁸ Microcracks begin developing continuously with the growth of the materials as caused by environmental factors including temperature, humidity, chemicals, and radiation.^{48,49} It creates a passageway for the entry of corrosive liquids like water, oxygen, and other substances, which makes thermo-mechanical properties of the materials deteriorated. Currently, the main techniques used to detect microcracks in materials are non-destructive testing (e.g., visual testing and computed tomography scanning).⁵⁰ These methods are often restricted by the discontinuity of the cracks, the depth of the defects, the shape of the materials itself, and technical barriers to the use of professional equipment.^{48,51} Therefore, it is crucial to develop straightforward and practical testing procedures.”

Additionally, in the previous version, we have made the following changes to these details:

“Comment 2”: Considering the limit of the number of words in the abstract, we have added corresponding contents in the revised abstract.

“Comment 5”: According to your good suggestion, we re-examined corresponding spectra, and these data have already been added in the Supplementary Information. We have also deleted the statement in the original manuscript and added a description of the data location in the Supplementary Information.

“Comment 6 and 7”: In the original manuscript, we mainly discussed the ratio of 700:1. Samples of other ratios are used as a comparison when participating in the discussions. Aiming at the problem of naming confusion, the following Supplementary Table 2 was added to the currently revised manuscript (Supplementary Table 2) and the description of currently revised manuscript (page 8) for Supplementary Table 4 was further simplified.

Supplementary Table 2. Naming of series 1 to 4 and different phosphorescence units.

VAc : Phosphor	2VN				1VN	9VA	MZ	NVP
	300:1	500:1	700:1	1000:1				
Polymerization (P)	P1	P2	P3	P4	1VN-P	9VA-P	MZ-P	NVP-P
Alcoholysis (H)	H1	H2	H3	H4	1VN-H	9VA-H	MZ-H	NVP-H
Cross-linking (B)	B1	B2	B3	B4	1VN-B	9VA-B	MZ-B	NVP-B

“Comment 8”: We have corrected the legend of Figure 2e from the previous manuscript, and the fluorescence information was listed in the Supplementary Table 3.

“Comment 9”: We explained the meaning of this paragraph in the response, and then rewrote

the description in the previous manuscript.

“Comment 10 and 11”: For the application, we explained why we chose these methods. For the detection of microcracks in the previous manuscript, some spectral data were added and discussed, the comparison of the scale was also added to the revised Figure 5a and the corresponding spectra are added to Supplementary Figure 53 from the previous Supplementary Information.

Please also note that, according to the policy from Nature Communications, the detailed response to reviewers' comments could be published together with the manuscript. Thus, readers can obtain more information. Many thanks again for your constructive comments to improve the quality of our manuscript.